# Artificial Intelligence Versus Professional Standards: A Cross-Sectional Comparative Study of GPT, Gemini, and ENT UK in Delivering Patient Information on ENT Conditions

**DOI:** 10.3390/diseases13090286

**Published:** 2025-09-01

**Authors:** Ali Alabdalhussein, Nehal Singhania, Shazaan Nadeem, Mohammed Talib, Derar Al-Domaidat, Ibrahim Jimoh, Waleed Khan, Manish Mair

**Affiliations:** 1Department of Otolaryngology, University Hospitals of Leicester, Leicester LE1 5WW, UK; n.singhania@nhs.net (N.S.); shazaan.nadeem@nhs.net (S.N.); mohammed.talib5@nhs.net (M.T.); derar.aldomaidat@nhs.net (D.A.-D.); waleed.a.khan@uhl-tr.nhs.uk (W.K.); 2Department of Maxillofacial Surgery, University Hospitals of Leicester, Leicester LE1 5WW, UK; ibrahim.jimoh1@nhs.net

**Keywords:** artificial intelligence, ENT, ChatGPT, Google Gemini

## Abstract

Objective: Patient information materials are sensitive and, if poorly written, can cause misunderstanding. This study evaluated and compared the readability, actionability, and quality of patient education materials on laryngology topics generated by ChatGPT, Google Gemini, and ENT UK. Methods: We obtained patient information from ENT UK and generated equivalent content with ChatGPT-4-turbo and Google Gemini 2.5 Pro for six laryngology conditions. We assessed readability (Flesch–Kincaid Grade Level, FKGL; Flesch Reading Ease, FRE), quality (DISCERN), and patient engagement (PEMAT-P for understandability and actionability). Statistical comparisons involved using ANOVA, Tukey’s HSD, and Kruskal–Wallis tests. Results: ENT UK showed the highest readability (FRE: 64.6 ± 8.4) and lowest grade level (FKGL: 7.4 ± 1.5), significantly better than that of ChatGPT (FRE: 38.8 ± 10.5, FKGL: 11.0 ± 1.5) and Gemini (FRE: 38.3 ± 8.5, FKGL: 11.9 ± 1.2) (all *p* < 0.001). DISCERN scores did not differ significantly (ENT UK: 21.3 ± 7.5, GPT: 24.7 ± 9.1, Gemini: 29.5 ± 4.6; *p* > 0.05). PEMAT-P understandability results were similar (ENT UK: 72.7 ± 8.3%, GPT: 79.1 ± 5.8%, Gemini: 78.5 ± 13.1%), except for lower GPT scores on vocal cord paralysis (*p* < 0.05). Actionability was also comparable (ENT UK: 46.7 ± 16.3%, GPT: 41.1 ± 24.0%, Gemini: 36.7 ± 19.7%). Conclusion: GPT and Gemini produce patient information of comparable quality and engagement to ENT UK but require higher reading levels and fall short of recommended literacy standards.

## 1. Introduction

Over recent decades, medical organisations have begun to provide information about medical conditions on online platforms. Since the late 1990s, medical organisations have increasingly provided health information online [1]. By the 2010s, the Mayo Clinic, a highly reputable organisation, had developed comprehensive, evidence-based content aimed at improving public understanding [2]. In the United Kingdom, the government began to centralise the source of information by launching NHS Conditions in 2007, a website that explains, at the patient level, most medical conditions [3,4]. However, recently, the ENT UK organisation tried to specifically address ENT conditions by launching the ENT UK website, which contains a variety of ENT conditions with an explanation at the patient level. Nonetheless, trust alone does not always dictate patient behaviour. In practice, many individuals still look beyond these official sites for additional perspectives or faster answers, despite the availability of trusted resources such as NHS Conditions and the ENT UK website. Multiple factors influence this shift. Barriers to in-person care, combined with the convenience of digital access, have encouraged more people to seek online guidance for immediate assistance. This trend reflects a broader increase in patients researching their health conditions online, both before and after consulting healthcare professionals. In recent years, this pattern has also extended to the use of AI-based tools for health-related queries [5]. The perception of online health information has resulted in a notable increase in physician visits, accompanied by a rise in online health information-seeking behaviours. Yet, as more people rely on online health information, there are real risks. The quality, accuracy, and readability of the information patients find can vary significantly, and many sources are not regulated or reviewed by professionals, which means people may not always receive safe or clear advice [6]. These concerns encouraged an academic evaluation of online content. Prior to the rise in Large Language Models (LLMs), studies of search engine responses (e.g., Yahoo, Google, Bing) showed variable and often suboptimal quality [7,8]. Such inconsistency risks misleading patients, driving anxiety, poor decisions, delayed care, and unnecessary consultations that increase costs [9,10].

### Artificial Intelligence as a Source of Patient Information

Artificial intelligence (AI) technologies, including language models such as ChatGPT and Gemini, are increasingly utilised by patients as accessible sources of medical information [11]. The widespread availability and user-friendly interfaces of these platforms, combined with their ability to generate rapid and coherent responses, have contributed to their fast growth [12]. As a result, patients are now more inclined to consult these models when seeking information about their symptoms or medical conditions, often before engaging with healthcare professionals [13]. Few studies have evaluated the effectiveness of AI language models, such as ChatGPT and Gemini, in producing patient information specific to otolaryngology [14]. Concerns about this are addressed in the work of Monteith et al. (published a study in The British Journal of Psychiatry in 2024). They noted that such content could contain factual errors, be incoherent, include false references, or potentially provide harmful advice [15]. Despite studies on AI content quality and readability, few have compared outputs directly with professional standards such as ENT UK. Additionally, the understandability and actionability of these materials remain insufficiently explored. This gap in the literature needs to be addressed, as patients increasingly rely on self-directed learning through the use of these technologies. As a result, systematic comparison is necessary to determine whether AI tools can safely complement or potentially replace traditional patient education resources in ENT practice. This is the first study to directly compare GPT-4o, Gemini 2.5 Pro, and ENT UK across multiple ENT subspecialties using four validated tools (FRE, FKGL, DISCERN, PEMAT-P). By combining diverse conditions, advanced AI systems, and a professional benchmark, we provide new evidence on whether AI-generated materials can match professional standards in patient education.

## 2. Materials and Methods

This study is a cross-sectional comparative study. We followed the STROBE (Strengthening the Reporting of Observational Studies in Epidemiology) checklist for cross-sectional studies [16]. The primary aim was to assess and compare the readability, quality, and actionability of patient education materials generated by two artificial intelligence (AI) models, OpenAI’s GPT-4o and Google’s Gemini 2.5 Pro, with the professional patient information provided by the ENT UK website.

We conducted the data collection process in May 2025. It was thorough and meticulous. We obtained AI-generated content using guest (non-logged-in) profiles to simulate the public user experience and to eliminate the effect of personalised algorithms. For each ENT condition, both GPT-4o and Gemini 2.5 Pro, we generated a response using a pre-designed, standardised template: “Create patient information content about [condition] that answers the following questions: What is [condition]? What are the symptoms of [condition]? What are the causes of [condition]? How is this condition diagnosed? What is the management?”

After obtaining the consent to use from ENT UK, we directly copied and pasted the responses from ENT UK into a draft file, and we anonymised all content to ensure a blinded analysis. The study included six selected ENT conditions: septal perforation, juvenile nasopharyngeal angiofibroma, otosclerosis, vestibular migraine, tracheomalacia in children, and thyroglossal cyst. The six conditions were chosen because they are well-described on the ENT UK website and cover all five standardised patient questions, enabling direct and fair comparison across sources. They also span different subspecialties within ENT, providing a diverse test set.

These three sources were selected to represent two of the most advanced and widely used large language models currently available to the public (GPT-4o and Gemini 2.5 Pro) and a professionally curated patient information source (ENT UK) that meets established standards in otolaryngology patient education. This combination allowed us to compare emerging AI-generated materials against a recognised professional benchmark.

The primary outcome measures were readability, quality of information, and actionability. We assessed readability using the Flesch Reading Ease (FRE) and Flesch–Kincaid Grade Level (FKGL) scores [17], both of which were calculated using Microsoft Word’s built-in readability tool. Higher Flesch Reading Ease (FRE) scores indicated greater ease of reading, while the Flesch-Kincaid Grade Level (FKGL) reflected the United States school grade level required to comprehend the text.

To assess the quality of treatment information, we used the DISCERN instrument [18]. Because guest-mode AI outputs do not provide citations or update metadata, we excluded DISCERN Section 1 (reliability) to avoid structural bias against AI related to formatting rather than content. We therefore scored only Sections 2–3 (treatment information and overall quality), aligning the assessment with our primary objective of comparing patient-facing content quality and usability across sources. We evaluated understandability and actionability using the Patient Education Materials Assessment Tool for Printable Materials (PEMAT-P) [19]. This tool was applied in accordance with official guidance. Six ENT surgeons independently reviewed and scored each document, blinded to the source of the content to minimise potential bias.

These tools (FRE, FKGL, DISCERN, PEMAT-P) were selected because they are validated, widely used in health communication research, and specifically suited to evaluating the readability, quality, and actionability of patient-facing educational materials. Their combined use provides a comprehensive assessment from linguistic, informational, and practical perspectives.

To ensure rigour and comparability, we took several steps to minimise bias. We used uniform prompting across AI platforms; all text outputs were anonymised prior to review, and the assessors were blinded to the content origin. The use of guest-mode queries eliminated the influence of personalised algorithms. Furthermore, the exclusion of DISCERN Section 1 ensured a fair assessment of the quality between human-authored and AI-generated material.

The final sample consisted of 18 responses (six for each of the six ENT conditions, with three sources per condition). This number was deemed sufficient to allow structured comparison while maintaining feasibility and depth of review.

ANOVA was applied to FRE and FKGL scores because these data met the assumptions for parametric testing, allowing comparison of mean values across the three sources. Kruskal–Wallis tests were used for DISCERN and PEMAT-P scores because these data did not meet the normality assumption, making a non-parametric approach more appropriate for comparing median values.

Registration: This study was prospectively registered on the Open Science Framework (OSF; DOI: https://doi.org/10.17605/OSF.IO/R9P4M; registration link: https://osf.io/r9p4m, accessed on 28 June 2025).

## 3. Results

### 3.1. Readability

A one-way ANOVA was conducted to compare the Flesch Reading Ease (FRE) scores among ENT UK, ChatGPT, and Gemini across six ENT conditions (Table 1). A statistically significant difference was found, F(2,15) = 16.16, *p* = 0.00018. Post hoc Tukey’s Honest Significant Difference (HSD) test showed that ENT UK produced significantly higher FRE scores (greater readability) than both ChatGPT (mean difference = 25.83, *p* < 0.05) and Gemini (mean difference = 26.33, *p* < 0.05), with no significant difference between ChatGPT and Gemini (mean difference = 0.50, *p* > 0.05) (Table 2, Figure 1).

For the Flesch–Kincaid Grade Level (FKGL) scores, ANOVA also revealed significant differences, F(2,15) = 17.33, *p* = 0.00013 (Table 3). ENT UK scored significantly lower (simpler grade level) than both ChatGPT (mean difference = 3.62, *p* < 0.05) and Gemini (mean difference = 4.53, *p* < 0.05). No significant difference was detected between ChatGPT and Gemini (mean difference = 0.92, *p* > 0.05) (Table 4, Figure 1).

Interpretation: ENT UK materials were easier to read and written at a lower grade level than both AI-generated sources.

### 3.2. Quality of the Material 

We evaluated DISCERN scores with one-way ANOVA and found no significant differences between the sources, F(2,15) = 1.90, *p* = 0.184 (Table 5). A Kruskal–Wallis test confirmed this result, H = 3.53, *p* = 0.172. Figure 2 shows overlapping distributions for ENT UK, ChatGPT, and Gemini.

Interpretation: All three sources delivered comparable quality in treatment-related information.

### 3.3. Understandability

We analysed PEMAT-P Understandability scores with one-way ANOVA and found no significant differences, F(2,15) = 1.90, *p* = 0.18 (Table 6). A Kruskal–Wallis test confirmed the result, H = 2.56, *p* = 0.278. Figure 3 displays overlapping medians and interquartile ranges for all sources.

Interpretation: ENT UK and AI-generated content provided equally clear and understandable information.

### 3.4. Actionability

We compared PEMAT-P Actionability scores using one-way ANOVA and found no significant differences, F(2,15) = 1.90, *p* = 0.18 (Table 7). A Kruskal–Wallis test supported this finding, H = 0.70, *p* = 0.704. Figure 4 illustrates the similar score distributions.

Interpretation: AI-generated and ENT UK materials guided patients toward actionable health behaviours to a similar extent.

### 3.5. Summarisation of the Results 

To summarise the quantitative findings across all metrics, the three charts provide a comparative analysis of mean scores and standard deviations for five key metrics: FRE, FKGL, DISCERN, PEMAT-P Understandability, and Actionability for ENT UK, GPT, and Gemini. ENT UK demonstrates the highest readability (FRE) and the lowest FKGL, indicating simpler language use with relatively low variability, making it the most consistent and accessible source. GPT shows moderate to high scores in content quality and understandability but exhibits high variability in DISCERN and Actionability, suggesting inconsistency across topics. Gemini performs similarly to GPT in average scores but with greater consistency in DISCERN and FKGL, indicating more stable quality and grade-level output. However, its PEMAT-P Actionability scores still display notable variability. Overall, ENT UK remains the most consistent and readable, while AI models, although comparable in quality, show greater variability in performance across different conditions (Figure 5, Figure 6, Figure 7 and Figure 8) and Table 8.

## 4. Discussion

The medical information generated by artificial intelligence has increased significantly [20]. With such exponential growth, there is an increasing demand to assess the readability, quality, and accessibility of this content. It is unclear whether AI-generated information is reliable and readable. It is still a matter of debate. Bibault et al. [21] and Lee et al. [22] have discussed that AI-generated responses are not comparable to those of physicians, particularly in terms of patient satisfaction. However, Nasra et al. [23] found in their review that AI can effectively simplify basic medical information, although balancing readability with the accuracy required for medical content still needs focused development.

It is essential to consider the readability of medical information, as this can impact both treatment decisions and patients’ psychological well-being [24,25].

The National Health System (NHS) in the UK reports that over 40% of adults struggle to comprehend publicly available health information, with more than 60% facing difficulties when content includes numerical data or statistics. This challenge arises largely because much health information is inadvertently written for individuals with higher levels of health literacy, making it less accessible to the general population [26]. To address these gaps, the Patient Information Forum (PIF) recommends using plain language, avoiding jargon and complex medical terms, and targeting a reading level suitable for 9- to 11-year-olds [27]. Additionally, Health Education England recommends that written patient information materials should be written at a level that can be understood by an 11-year-old [28].

In our study, we found that readability was better on the professional website (ENT UK) compared to AI-generated content from GPT and Gemini. This outcome can be attributed to several factors. First, readability levels continue to favour human-written material. A 2023 study published in Cureus compared AI-generated patient education with physician-authored content, finding that AI outputs frequently exceeded recommended readability levels, making them harder for patients to understand. In contrast, human-written texts were more likely to align with the recommendations of the American Medical Association (AMA) and the National Institutes of Health (NIH) for patient materials, specifically within the 6th- to 8th-grade reading range [29,30].

Second, challenges in cultural competence persist. A scoping review published in 2024 examined the role of AI in crafting culturally sensitive public health messages, noting that while AI can incorporate cultural nuances, its effectiveness is often limited by low user acceptance, ethical concerns, and a general lack of trust in AI-generated content [31]. Third, AI-generated materials frequently lack the clinical judgement and contextual prioritisation inherent to human-authored content. Unlike clinicians, large language models such as GPT and Gemini may produce factually accurate text but struggle to triage and emphasise the most critical information, particularly across diverse cultural or situational contexts. This shortfall can undermine the clarity and practical relevance of patient education materials [31].

Regarding quality, understandability, and actionability, we did not find significant statistical differences between patient information generated by the AI models and the professional standards of ENT UK. When evaluated using DISCERN, all three sources scored similarly, suggesting that AI-generated materials are beginning to match the standards set by professional organisations in terms of accuracy, objectivity, and depth. Likewise, PEMAT-P scores for understandability indicated that both AI models produced content that was generally well-structured, easy to comprehend, and written in accessible language, performing comparably to ENT UK. Notably, there were also no discernible differences in actionability, indicating that treatment-related information from AI was equally effective in guiding patients toward specific health actions or decisions.

While human-written materials remain superior in readability, the overall parity in quality and engagement-related measures suggests a growing potential for large language models to support patient education. However, these tools should be carefully vetted and contextualised by healthcare professionals to ensure they meet readability standards and address patient needs safely and effectively.

## 5. Limitation

The following points can summarise this study’s limitations: First, our template was based on ENT UK’s available questions; these resources use formal language and reflect the view of healthcare professionals, which may not align with how patients typically search for information online. Patients are more likely to type informal, symptom-based questions such as “Why is my throat hoarse in the morning?” rather than clinical terms like “What is vocal cord paralysis?” As a result, the AI outputs may not accurately reflect how the models would respond to real-world patient queries. 

Second, we did not design an advanced AI-questions template. For example, we did not ask it to tailor its responses by adjusting the format, length, or reading level. This lack of prompt customisation influenced the quality and clarity of the AI’s responses. Third, all the assessments performed by a small group of reviewers with varying ENT experience (junior to higher speciality levels) could have introduced some bias into the evaluation process. Finally Our study did not assess the clinical accuracy of the AI-generated content, which remains essential for patient safety. These limitations have practical implications. Using formal, professional wording may underestimate how AI performs in typical patient-led searches, potentially misrepresenting its accessibility in real-world contexts. Similarly, the absence of tailored prompts means our findings reflect baseline AI performance rather than its optimised potential, which may be higher. A broader agreement would also be necessary due to Reviewer variability to enhance reliability, and the absence of clinical accuracy assessment raises safety concerns, as even highly readable content could spread misinformation if unverified. Future work should include expert validation and involve patients directly to evaluate real-world clarity and trust.

## 6. Future Applications

Looking to the future, this project demonstrates how AI can play a larger role in patient communication, particularly in ENT care. As tools like GPT and Gemini become more sophisticated, there is real potential to use them to generate clear, personalised information for patients that matches their level of understanding. With proper guidance from a professional clinician, AI-generated content could improve how medical conditions are explained to patients. In the future, we see AI supporting NHS services by helping patients prepare for appointments or understand their diagnosis and treatment options afterwards. One suggestion is to combine the strengths of professional standards with the flexibility of AI models (for example, a chatbot) to produce more accurate and easy-to-read resources. 

Future studies could combine expert checks on clinical accuracy with patient feedback, helping to ensure AI-generated materials are both safe and truly patient-friendly. Establishing clear standards could also guide the responsible integration of these tools into clinical practice.

## 7. Conclusions

This study demonstrates that while GPT and Gemini can produce patient information of comparable quality to ENT UK standards, their content is less readable and does not adhere to recommended literacy levels. Both patients and clinicians need to ensure that AI-generated materials are not only accurate but also accessible and understandable. We recommend that professional standards guide the further development of AI-generated content.

For healthcare providers, it remains highly advisable to rely on professional patient leaflet information, and it would be premature to recommend patients start using AI-generated content without direct supervision. Incorporating AI tools into patient education should serve as a supplement to, rather than a replacement for, clinician–patient communication, with continuous monitoring of AI content quality and relevance.

## Figures and Tables

**Figure 1 diseases-13-00286-f001:**
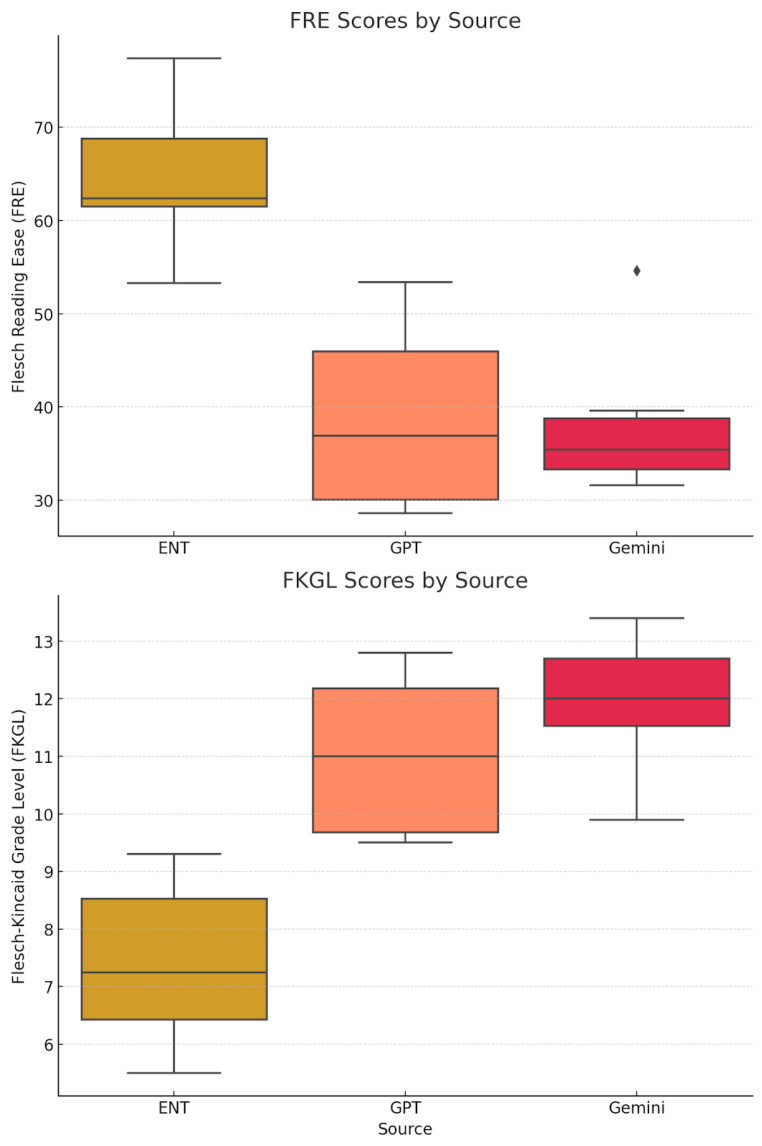
Distribution of Flesch Reading Ease (FRE) and Flesch–Kincaid Grade Level (FKGL) Scores by Source.

**Figure 2 diseases-13-00286-f002:**
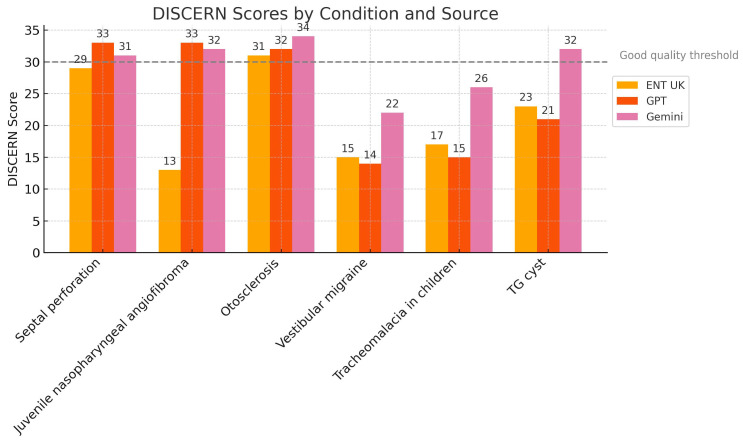
Distribution of DISCERN Scores by Information Source.

**Figure 3 diseases-13-00286-f003:**
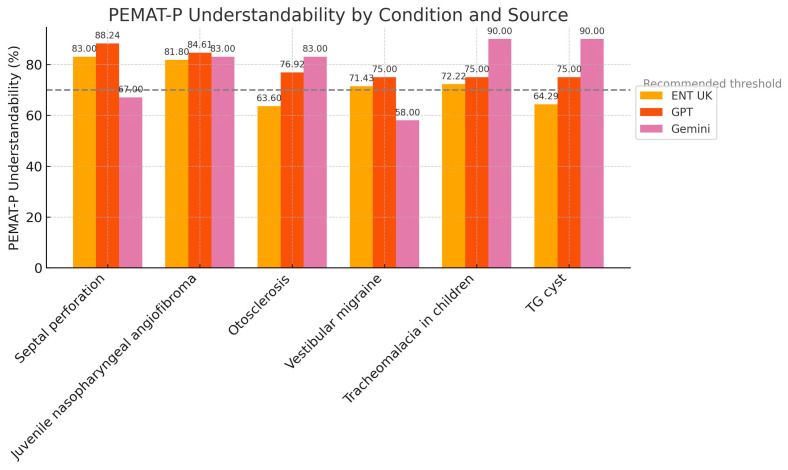
Distribution of PEMAT-P Understandability Scores by source.

**Figure 4 diseases-13-00286-f004:**
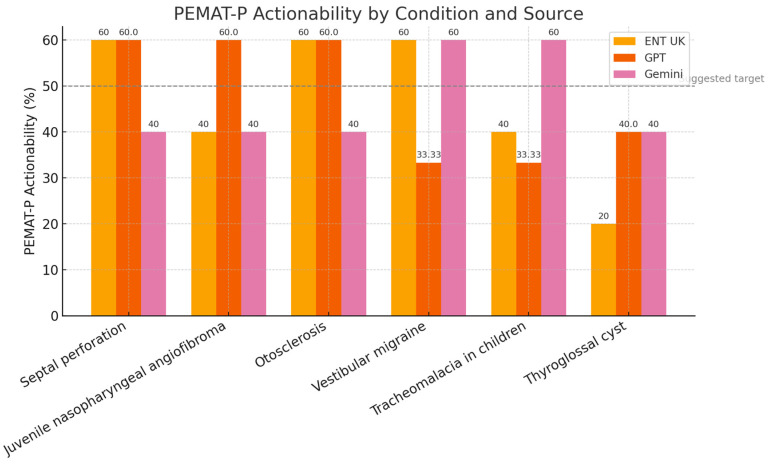
Distribution of PEMAT-P Actionability Scores by Source.

**Figure 5 diseases-13-00286-f005:**
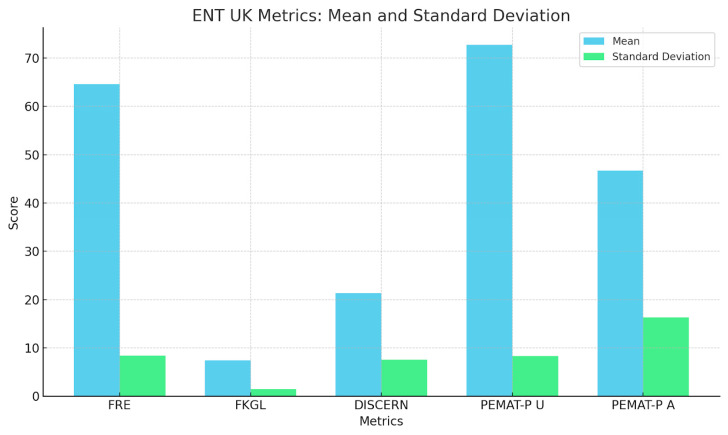
Mean and Standard Deviation of ENT UK Patient Information Metrics.

**Figure 6 diseases-13-00286-f006:**
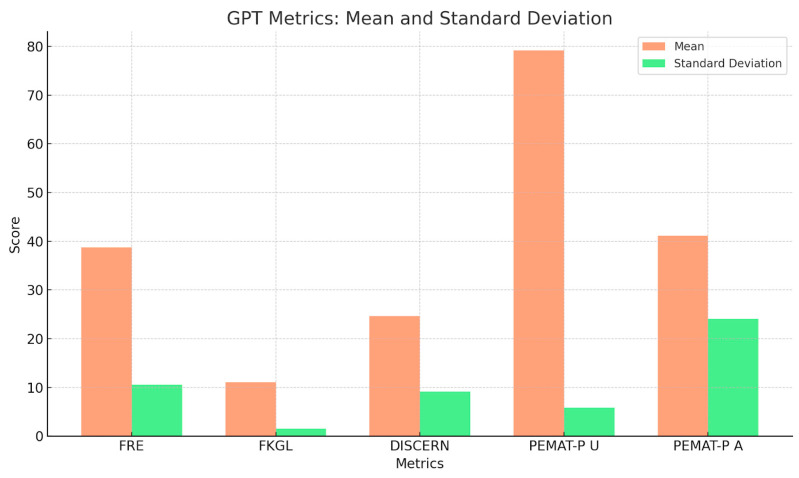
Mean and Standard Deviation of GPT-Generated Patient Information Metrics.

**Figure 7 diseases-13-00286-f007:**
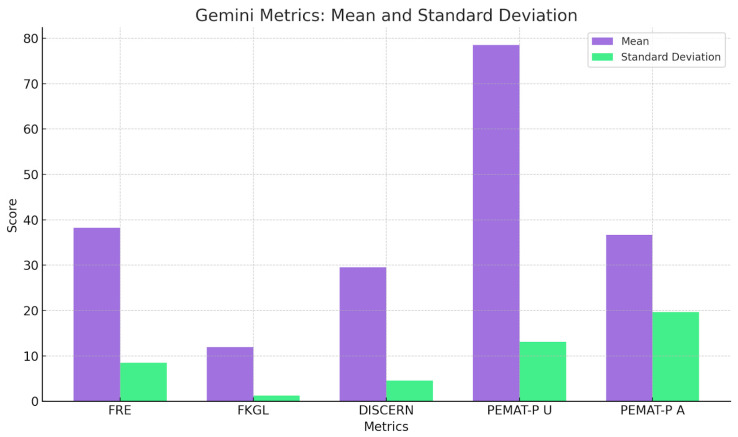
Mean and Standard Deviation of Gemini-Generated Patient Information Metrics.

**Figure 8 diseases-13-00286-f008:**
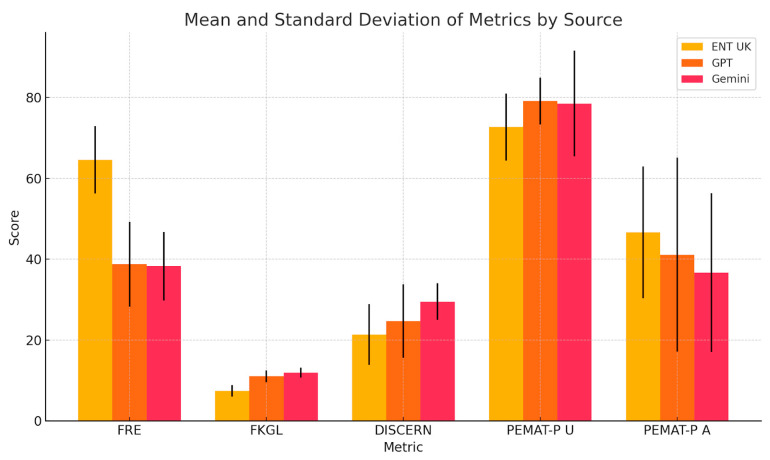
Comparative Bar Chart of Mean Scores and Standard Deviations Across Metrics for ENT UK, GPT, and Gemini.

**Table 1 diseases-13-00286-t001:** One-way ANOVA of the Flesch Reading Ease (FRE) scores among ENT UK, ChatGPT, and Gemini.

Source of Variation	SS	DF	MS	F	*p* Value	F Crit
Between Groups	2722.11	2	1361.05	16.16	0.00018	3.68
Within Groups	1263.19	15	84.2132			
Total	3985.30	17				

SS: Sum of Squares, df: Degrees of Freedom, MS: Mean Square, F: F-ratio (or F-statistic), *p*-value: Probability value, F crit: Critical value of F.

**Table 2 diseases-13-00286-t002:** Post hoc analysis of the Flesch Reading Ease (FRE) scores using Tukey’s Honest Significant Difference (HSD) test.

Pairwise FRE Comparison	Mean Difference	HSD Threshold	Significant?
ENT UK vs. ChatGPT	25.83	13.76	Yes
ENT UK vs. Gemini	26.33	13.76	Yes
ChatGPT vs. Gemini	0.5	13.76	No

HSD: Honest Significant Difference.

**Table 3 diseases-13-00286-t003:** One-way ANOVA of the Flesch-Kincaid Grade Level (FKGL) scores among ENT UK, ChatGPT, and Gemini.

Source of Variation	SS	DF	MS	F	*p* Value	F Crit
Between Groups	68.94	2	34.47	17.32	0.00012	3.68
Within Groups	29.84	15	1.98			
Total	98.785	17				

SS: Sum of Squares, df: Degrees of Freedom, MS: Mean Square, F: F-ratio (or F-statistic), *p*-value: Probability value, F crit: Critical value of F.

**Table 4 diseases-13-00286-t004:** Post hoc analysis of the Flesch-Kincaid Grade Level (FKGL) score using Tukey’s Honest Significant Difference (HSD) test.

Pairwise FRE Comparison	Mean Difference	HSD Threshold	Significant?
ENT UK vs. ChatGPT	3.62	2.12	Yes
ENT UK vs. Gemini	4.53	2.12	Yes
ChatGPT vs. Gemini	0.92	2.12	No

HSD: Honest Significant Difference.

**Table 5 diseases-13-00286-t005:** Comparison of DISCERN Scores Across GPT, Gemini, and ENT UK Using One-Way ANOVA.

Source of Variation	SS	DF	MS	F	*p* Value	F Crit
Between Groups	202.33	2	101.16	1.89	0.18	3.68
Within Groups	800.16	15	53.34			
Total	1002.5	17				

SS: Sum of Squares, df: Degrees of Freedom, MS: Mean Square, F: F-ratio (or F-statistic), *p*-value: Probability value, F crit: Critical value of F.

**Table 6 diseases-13-00286-t006:** One-Way ANOVA Summary for PEMAT-P Understandability Scores Across ENT UK, GPT, and Gemini.

Source of Variation	SS	DF	MS	F	*p* Value	F Crit
Between Groups	202.33	2.00	101.17	1.90	0.18	3.68
Within Groups	800.17	15.00	53.34			
Total	1002.50	17.00				

SS: Sum of Squares, df: Degrees of Freedom, MS: Mean Square, F: F-ratio (or F-statistic), *p*-value: Probability value, F crit: Critical value of F.

**Table 7 diseases-13-00286-t007:** One-Way ANOVA Summary for PEMAT-A Actionability Scores Across ENT UK, GPT, and Gemini.

Source of Variation	SS	DF	MS	F	*p* Value	F Crit
Between Groups	202.33	2.00	101.17	1.90	0.18	3.68
Within Groups	800.17	15.00	53.34			
Total	1002.5	17.00				

SS: Sum of Squares, df: Degrees of Freedom, MS: Mean Square, F: F-ratio (or F-statistic), *p*-value: Probability value, F crit: Critical value of F.

**Table 8 diseases-13-00286-t008:** Mean and Standard Deviation of Readability, Quality, and Engagement Metrics for ENT UK, GPT, and Gemini.

Metric	ENT UK Mean	ENT UK SD	GPT Mean	GPT SD	Gemini Mean	Gemini SD
FRE	64.58	8.38	38.75	10.51	38.25	8.48
FKGL	7.4	1.48	11.02	1.5	11.93	1.23
DISCERN	21.33	7.53	24.67	9.09	29.5	4.55
PEMAT-P U	72.72	8.3	79.13	5.82	78.5	13.1
PEMAT-P A	46.67	16.33	41.11	24.01	36.67	19.66

SD: standard deviation.

## Data Availability

The data can be provided upon request.

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
