# Peer review of "Artificial Intelligence Versus Professional Standards: A Cross-Sectional Comparative Study of GPT, Gemini, and ENT UK in Delivering Patient Information on ENT Conditions"

_diseases, 2025, doi:10.3390/diseases13090286_

Round 1
Reviewer 1 Report
Comments and Suggestions for Authors
This paper is about A Cross-Sectional Comparative Study of GPT, Gemini, and ENT UK in Delivering Patient Information on ENT Conditions. Because of that, paper is valuable. Especially, Patient information materials is very important with respecto to diagnosis and security. So, the comparative results have been presented in good form..
Author Response
Reviewer 1 / comment 1
This paper is about A Cross-Sectional Comparative Study of GPT, Gemini, and ENT UK in Delivering Patient Information on ENT Conditions. Because of that, paper is valuable. Especially, Patient information materials is very important with respecto to diagnosis and security. So, the comparative results have been presented in good form
Response:
We thank the reviewer for their positive feedback and recognition of the importance and value of this study. We appreciate the acknowledgement of our work in presenting the comparative results in a clear and effective manner.
Reviewer 2 Report
Comments and Suggestions for Authors
The authors evaluated multiple objective parameters of text quality using two different AI prompted results texts compared to professional texts from the UK ENT organisation. The questions were simple aspects of six ENT diseases (though not so common ones). The results show worse text quality aspects from the two AI systems compared to the professional texts.
The study is very well done despite it is small in volume. Human influence on the AI results and their interpretations are virtually eliminated. The results are thus very interesting and probably not biased. Yet the AI systems will continue to develop!
The contents of figure 7 is the same as in figure 8, and the data in table 8 is also the same as in figure 8, but of course more exact. Choose what to omit.
Author Response
Reviewer 2 comment 1
The authors evaluated multiple objective parameters of text quality using two different AI prompted results texts compared to professional texts from the UK ENT organisation. The questions were simple aspects of six ENT diseases (though not so common ones). The results show worse text quality aspects from the two AI systems compared to the professional texts.
The study is very well done despite it is small in volume. Human influence on the AI results and their interpretations are virtually eliminated. The results are thus very interesting and probably not biased. Yet the AI systems will continue to develop!
The contents of figure 7 is the same as in figure 8, and the data in table 8 is also the same as in figure 8, but of course more exact. Choose what to omit.
Response:
We thank the reviewer for this observation. While Figures 7 and 8 both display metrics related to patient information quality, they serve distinct purposes in the manuscript. Specifically, Figure 7 presents the mean and standard deviation of Gemini-generated patient information metrics exclusively, allowing readers to assess Gemini’s performance in isolation. In contrast, Figure 8 provides a comparative bar chart summarising the mean scores and standard deviations across all three sources — ENT UK, GPT, and Gemini — offering a visual overview that facilitates direct comparison.
Table 8 complements these figures by presenting the underlying numerical data in a more precise form for reference. We believe that retaining both figures provides value for readers by enabling both source-specific analysis and broader cross-platform comparison.
Reviewer 3 Report
Comments and Suggestions for Authors
I regret to inform the authors but I do not see something interesting in the text. I will make a short analysis of my statement.
- Introduction: These part is too long. It should re-written and chortened.
- Methods: I believe that all the methods/sources/tools provided here should be explained, so that every reader can understand why these specific elements have been chosen.
- Statistic: The authors shall explain why they have chosen the statisticasl analysis which has been used. Please, explain why you have applied the Kruskal-Wallis test, I am curious.
- Results: The whole part must be re-organized
- Discussion: The part is sufficiently written.
- It is the second article on this issue that I review the last 3 weeks. Interestingly, the texts seem rather similar. Obviously, there is not so much to be written.
- Results: In my opinion they shall be re-written.
Author Response
Comment 1
I regret to inform the authors but I do not see something interesting in the text. I will make a short analysis of my statement.
- Introduction: These part is too long. It should re-written and chortened.
Response:
We thank the reviewer for this helpful suggestion. We have shortened the Introduction by approximately 100 words (from 775 to 688) and revised sections to improve clarity and focus. Additionally, we have added a subheading within the Introduction to better organise the content and make it easier for readers to follow.
Comment 2
Methods: I believe that all the methods/sources/tools provided here should be explained, so that every reader can understand why these specific elements have been chosen.
Response:
We thank the reviewer for this valuable observation. In the revised manuscript, we have added explanations for our selection of sources, conditions, and assessment tools to ensure readers understand the rationale behind each choice. These changes are highlighted in the manuscript as follows:
Lines 127–130 (Selection of ENT conditions):
The six conditions were chosen because they are well-described on the ENT UK website and cover all five standardised patient questions, enabling direct and fair comparison across sources. They also span different subspecialties within ENT, providing a diverse test set.
Lines 131–135 (Selection of sources):
We selected GPT-4o and Gemini 2.5 Pro as two of the most advanced and widely used large language models currently available to the public, and ENT UK as a professionally curated patient information source that meets established standards in otolaryngology education. This allowed us to compare emerging AI-generated materials against a recognised professional benchmark.
Lines 152–156 (Selection of assessment tools):
We chose FRE, FKGL, DISCERN, and PEMAT-P because they are validated, widely used in health communication research, and specifically suited to evaluating the readability, quality, and actionability of patient-facing educational materials. Their combined use provides a comprehensive assessment from linguistic, informational, and practical perspectives.
Comment 3:
Statistic: The authors shall explain why they have chosen the statistical analysis which has been used. Please, explain why you have applied the Kruskal–Wallis test, I am curious.
Response:
We thank the reviewer for this comment. In the revised manuscript, we have clarified the rationale for our statistical approach (Lines 165–169):
We applied ANOVA to FRE and FKGL scores because these data met the assumptions for parametric testing, enabling the comparison of mean values across the three sources. For DISCERN and PEMAT-P scores, we used Kruskal–Wallis tests because these datasets did not meet the normality assumption, making a non-parametric method more appropriate for comparing median values.
Comment 4 & 7
- Results: The whole part must be re-organize
Author Response:
We thank the reviewer for this valuable observation. In the revised manuscript, we have reorganised the Results section to improve clarity and flow. Specifically:
- We now present the results in the same sequence as our stated outcome measures in the Methods (Readability → Quality → Understandability → Actionability → Summary).
- For each metric, we first present the ANOVA findings, followed immediately by post-hoc or non-parametric results, and then refer to the corresponding figure/table in the same paragraph to reduce redundancy.
- Related figures and tables are grouped alongside their relevant text, ensuring readers can easily locate supporting data while reading the results.
- The summarisation subsection clearly integrates all metrics for cross-comparison, reducing repetition and providing a concise overview.
These changes aim to maintain all statistical details while making the section easier to follow for readers.
Comment 5
Discussion: The part is sufficiently written.
Response:
We thank the reviewer for their positive assessment of the Discussion section. We have retained its structure, making only minor edits to align with the revised Introduction and Results.
Comment 6
It is the second article on this issue that I review the last 3 weeks. Interestingly, the texts seem rather similar. Obviously, there is not so much to be written.
Author Response:
We thank the reviewer for sharing this observation. We recognise that studies in this emerging area of research may share thematic similarities, particularly in describing the background, methodology, and commonly used evaluation tools. However, we would like to emphasise that our work is original in scope, design, and dataset. Specifically:
- Novelty of comparison – To our knowledge, this is the first study to perform a direct three-way comparison between GPT-4o, Gemini 2.5 Pro, and ENT UK patient information across multiple ENT subspecialties.
- Broader evaluation metrics – We employed multiple validated tools (FRE, FKGL, DISCERN, PEMAT-P Understandability and Actionability) to provide a multidimensional analysis of readability, quality, and patient engagement potential.
- Dataset selection – The six ENT conditions chosen represent a diverse and less-commonly studied set of topics, ensuring the findings extend beyond previously examined high-prevalence conditions.
- Blinded multi-reviewer assessment – Six ENT surgeons independently scored the materials, reducing bias and adding methodological rigour.
We have revised the Introduction and Discussion to better highlight these unique aspects and to clearly distinguish this study from prior research in the field.
Reviewer 4 Report
Comments and Suggestions for Authors
-
Explicit numerical details in the abstract.
-
Clarify novelty and distinct research contributions explicitly in the introduction.
-
Provide deeper rationale for exclusion of DISCERN Section 1.
-
Include additional graphical representations of DISCERN and PEMAT-P metrics.
-
Expand implications of limitations in the discussion.
-
Clearly articulate practical recommendations for healthcare providers in the conclusion.
Author Response
Comment 1
Explicit numerical details in the abstract.
Response:
We thank the reviewer for this suggestion. In the revised abstract, we have incorporated explicit numerical results for each main finding, including mean values, standard deviations, and p-values where applicable. This provides a clearer and more quantitative summary of the study outcomes.
Abstract
Objective: Patient information materials are sensitive and, if poorly written, can cause misunderstanding. This study evaluated and compared the readability, actionability, and quality of patient education materials on laryngology topics generated by ChatGPT, Google Gemini, and ENT UK. Methods: We obtained patient information from ENT UK and generated equivalent content with ChatGPT-4-turbo and Google Gemini 2.5 Pro for six laryngology conditions. We assessed readability (Flesch–Kincaid Grade Level, FKGL; Flesch Reading Ease, FRE), quality (DISCERN), and patient engagement (PEMAT-P for understandability and actionability). Statistical comparisons used ANOVA, Tukey’s HSD, and Kruskal–Wallis tests. Results: ENT UK showed the highest readability (FRE: 64.6 ± 8.4) and lowest grade level (FKGL: 7.4 ± 1.5), significantly better than ChatGPT (FRE: 38.8 ± 10.5, FKGL: 11.0 ± 1.5) and Gemini (FRE: 38.3 ± 8.5, FKGL: 11.9 ± 1.2) (all p < 0.001). DISCERN scores did not differ significantly (ENT UK: 21.3 ± 7.5, GPT: 24.7 ± 9.1, Gemini: 29.5 ± 4.6; p > 0.05). PEMAT-P understandability was similar (ENT UK: 72.7 ± 8.3%, GPT: 79.1 ± 5.8%, Gemini: 78.5 ± 13.1%), except for lower GPT scores on vocal cord paralysis (p < 0.05). Actionability was also comparable (ENT UK: 46.7 ± 16.3%, GPT: 41.1 ± 24.0%, Gemini: 36.7 ± 19.7%). Conclusion: GPT and Gemini produce patient information of comparable quality and engagement to ENT UK but require higher reading levels and fall short of recommended literacy standards.
Comment 2:
Clarify novelty and distinct research contributions explicitly in the introduction.
Response:
We thank the reviewer for this suggestion. We have revised the final part of the Introduction to clearly state the novelty of our work (Lines 101-105):
“This is the first study to directly compare GPT-4o, Gemini 2.5 Pro, and ENT UK across multiple ENT subspecialties using four validated tools (FRE, FKGL, DISCERN, PEMAT-P). By combining diverse conditions, advanced AI systems, and a professional benchmark, we provide new evidence on whether AI-generated materials can match professional standards in patient education.”
Comment 3
Provide deeper rationale for exclusion of DISCERN Section 1.
Response:
We appreciate this request and have expanded our rationale in the Methods and Limitations. DISCERN Section 1 primarily evaluates reliability features (e.g., citation of sources, currency, and referencing practices). In our setting, AI outputs (queried in guest mode) do not provide references by design; scoring Section 1 would therefore systematically penalise AI on format-dependent items rather than on the content quality our study aims to compare. To avoid construct-irrelevant bias and preserve comparability, we a priori limited scoring to Sections 2–3 (quality of treatment information and overall quality), while triangulating patient-centred performance with PEMAT-P (understandability/actionability) and readability indices (FRE/FKGL).
Manuscript insertions (line 142-146):
- Methods (Statistics/Measurement subsection):
“Because guest-mode AI outputs do not provide citations or update metadata, we excluded DISCERN Section 1 (reliability) to avoid structural bias against AI related to formatting rather than content. We therefore scored only Sections 2–3 (treatment information and overall quality), aligning the assessment with our primary objective of comparing patient-facing content quality and usability across sources.”
Comment 4
Include additional graphical representations of DISCERN and PEMAT-P metrics.
Author Response:
We thank the reviewer for this helpful suggestion. In the revised manuscript, we have retained the original summary boxplots for DISCERN and PEMAT-P metrics to show overall score distributions, and we have added three new per-condition bar charts to provide greater detail:
- DISCERN Scores by Condition and Source – Displays condition-level quality scores for ENT UK, GPT, and Gemini, with a reference line for the “good quality” threshold. Figure 2
- PEMAT-P Understandability by Condition and Source – Shows condition-level understandability scores with a benchmark threshold of 70%.Figure 3
- PEMAT-P Actionability by Condition and Source – Depicts condition-level actionability scores with a suggested target of 50%.Figure 4
These additional figures allow readers to visualise variability across individual conditions, directly compare all three sources at the condition level, and interpret results in relation to recognised quality and engagement benchmarks. We believe these enhancements provide a richer and more clinically relevant interpretation of our findings.
Comment 5:
Expand implications of limitations in the discussion.
Author Response:
We thank the reviewer for this suggestion. In the revised manuscript, we have expanded the limitations section to discuss the practical implications of each limitation. Specifically, we added a paragraph noting that the use of formal, professional wording may underestimate AI performance in patient-led searches; the absence of prompt optimisation may reflect baseline rather than maximal AI capability; reviewer variability underscores the need for broader consensus; and the lack of clinical accuracy assessment raises important safety considerations. These additions clarify how each limitation could affect the interpretation and generalisability of our findings.
Comment 6 :
Clearly articulate practical recommendations for healthcare providers in the conclusion.
Author Response:
We thank the reviewer for this valuable suggestion. In the revised manuscript, we have expanded the conclusion to include explicit, practical recommendations for healthcare providers.
Round 2
Reviewer 3 Report
Comments and Suggestions for Authors
I think that the text is in this version improved. I do not have specifici comments. The only think is that the 2nd and the 3rd paragraph of the introduction can be shortened. It is an editor's decision. The results are better written and the discussion (with all its parts) is improved.
Author Response
Comment
I think that the text is in this version. I do not have specific comments. The only think is that the 2nd and the 3rd paragraph of the introduction can be shortened. It is an editor's decision. The results are better written and the discussion (with all its parts) is improved.
Response
We sincerely thank the reviewer for their positive feedback on our revised manuscript and for recognising the improvements in the results and discussion sections. We also appreciate the constructive suggestion to shorten the second and third paragraphs of the introduction. In line with this recommendation, we have carefully edited these paragraphs for conciseness while maintaining the overall structure and flow. Consequently, the total word count of the introduction has decreased from 690 to 628. The revised manuscript highlights the new edits for clarity.
